# Enhancing the Consistency and Performance of Graphene-Based Devices via Al Intermediate-Layer-Assisted Transfer and Patterning

**DOI:** 10.3390/nano14070568

**Published:** 2024-03-25

**Authors:** Yinjie Wang, Ningning Su, Shengsheng Wei, Junqiang Wang, Mengwei Li

**Affiliations:** 1Academy for Advanced Interdisciplinary Research, North University of China, Taiyuan 030051, China; wongyinchieh@163.com; 2State Key Laboratory of Dynamic Measurement Technology, North University of China, Taiyuan 030051, China; suningning826032@163.com (N.S.); 20230137@nuc.edu.cn (S.W.); 3School of Semiconductors and Physics, North University of China, Taiyuan 030051, China

**Keywords:** graphene transfer, graphene patterning, Al intermediate layer, polymer residue, graphene temperature sensor

## Abstract

Graphene has garnered widespread attention, and its use is being explored for various electronic devices due to its exceptional material properties. However, the use of polymers (PMMA, photoresists, etc.) during graphene transfer and patterning processes inevitably leaves residues on graphene surface, which can decrease the performance and yield of graphene-based devices. This paper proposes a new transfer and patterning process that utilizes an Al intermediate layer to separate graphene from polymers. Through DFT calculations, the binding energy of graphene–Al was found to be only −0.48 eV, much lower than that of PMMA and photoresist with graphene, making it easier to remove Al from graphene. Subsequently, this was confirmed through XPS analysis. A morphological characterization demonstrated that the graphene patterns prepared using the Al intermediate layer process exhibited higher surface quality, with significantly reduced roughness. It is noteworthy that the devices obtained with the proposed method exhibited a notable enhancement in both consistency and sensitivity during electrical testing (increase of 67.14% in temperature sensitivity). The low-cost and pollution-free graphene-processing method proposed in this study will facilitate the further commercialization of graphene-based devices.

## 1. Introduction

Since graphene was first discovered in experiments at the University of Manchester in 2004 [1], its extraordinary physical properties have aroused great interest among researchers. By virtue of the superior mechanical stability [2], ultrahigh carrier mobility [3], unique adsorption capability [4], and other characteristics of graphene, a series of innovative electronic devices have been developed and demonstrate significant potential application in the fields of optoelectronics, sensor systems, micro-energy [5], etc. However, this also presents greater demands or challenges to ensure the appropriate surface quality of graphene. Research indicated that pollution, wrinkles, cracks, and other factors on the surface of graphene substantially impact a graphene film’s electrical properties, leading to a reduction in electrical conductivity and carrier mobility, as well as to a lack of consistency [6,7,8]. Among contaminants, polymethyl methacrylate (PMMA) and photoresists serve as significant contributors to the residual pollution found in graphene [9].

Hence, devising strategies to mitigate the deposition of these polymers during the graphene transfer and patterning processes becomes pivotal in enhancing the performance and yield of electronic devices based on graphene. For this reason, researchers have put forth a variety of approaches, including chloroform washing [10], thermal annealing [11], and plasma cleaning [12], to eliminate organic contaminants from the surface of graphene. Nevertheless, it should be noted that these methods come with considerable drawbacks. For instance, chloroform poses high toxicity risks, thermal annealing falls short in completely eliminating organic matter and may result in graphene oxidation [13], while plasma cleaning can introduce new defects or even degrade the quality of graphene [14].

Addressing the challenge of polymer removal, researchers introduced a novel transfer method involving the insertion of a metal layer between the polymer and graphene to create a barrier, effectively isolating graphene from the polymer [15,16]. Yamujin Jang et al. achieved successful graphene transfer through the utilization of Au as an intermediate dielectric layer, ensuring an ultra-clean transfer process [17]. Additionally, the Au intermediate layer effectively served as an isolator for the employed photoresist during the subsequent patterning steps. Nevertheless, Au presents significant drawbacks when used as an intermediate layer. For instance, the removal of the Au intermediate layer necessitates the utilization of a potent oxidizing Au-etching solution (1:4:40 I_2_/KI/H_2_O), which can potentially lead to the oxidation of graphene. Moreover, the Au mask pattern is obtained through wet etching, resulting in suboptimal pattern accuracy. Hence, in order to attain graphene patterns of superior quality, further investigation of this method is warranted.

Here, we propose the utilization of Al as the material for the intermediate layer, effectively addressing the limitations of the aforementioned transfer method, due to the cost-effectiveness of the Al material and the possibility of achieving a high-precision intermediate layer pattern through dry etching with Cl_2_ and BCl_3_ [18]. In addition, the Al intermediate layer can be removed using a commonly used developer solution, i.e., a 2.38% tetramethylammonium hydroxide solution (3038), which is safe for most materials such as Si, SiO_2_, and SiNx. In this work, we employ the H_2_ bubbling transfer method and a specialized developer solution to effectively handle the issue of Al susceptibility to corrosion by the used Cu etchant and developer during the transfer and development processes.

## 2. Materials and Methods

In this study, monolayer graphene (grown via chemical vapor deposition on a copper substrate) purchased from the Beijing Graphene Institute in Beijing, China was used. Furthermore, the target substrate for transfer, which was prepared using a Si thermal oxidation process to create SiO_2_, had metal electrodes (Au/Cr) deposited on it beforehand via magnetron sputtering. In this process, the graphene samples used were cut from the same piece of copper foil, and the target substrates for transfer were also prepared from the same batch to avoid the influence of extraneous factors.

### 2.1. Transfer Process of Graphene onto SiO_2_/Si Substrates

The transfer process initially involved separating graphene from the copper substrate using a bubble-assisted method [19] with Na_2_SO_4_ as the electrolyte. Subsequently, graphene was transferred onto a thermal release tape using the polydimethylsiloxane (PDMS) dry transfer method [20]. To compare the processing methods using the Al intermediate layer with the traditional process with PMMA and a photoresist in direct contact with graphene, two types of samples were prepared. The process flow for the deposition of the Al intermediate layer is shown in Figure 1. Firstly, the Al intermediate layer (50 nm) was deposited onto graphene surface using electron beam evaporation at a low rate (~2 Å/s) to avoid damage to graphene. Subsequently, a PMMA support layer was spin-coated onto the Al layer surface. Then, the Cu substrate in this stack was pasted onto a hard adhesive tape, and a layer of PDMS (200 μm) was laminated on the surface of PMMA (Figure 1a). Subsequently, the sample was immersed in a NaSO_4_ solution for bubble separation, as shown in Figure 1b. The PDMS/PMMA/Al/Gr/Cu/hard tape stack was positioned at the cathode, and due to cathodic current protection, Al was not corroded (Appendix A). In addition, the hard tape on the Cu substrate reduced unnecessary reaction areas and improved the bubble separation rate. After the separation, the graphene stack was rinsed in deionized water, dried, and then laminated onto a SiO_2_/Si substrate using a press, (Figure 1c). The PDMS film was then separated by heating the sample, as shown in Figure 1d. Finally, acetone was used to remove PMMA, resulting in the Al/Gr structure. The samples prepared by this method were labeled Al-Gr (Al–graphene). In comparison, the control group samples prepared using the traditional method did not have an Al intermediate layer deposited on graphene. PMMA was directly in contact with graphene, and the subsequent bubble transfer process was the same as that for Al-Gr. The control group samples were labeled TP-Gr (Traditional Process–Graphene). The samples after transfer using the two methods are shown in Figure 1e.

### 2.2. Fabrication Process of Graphene Patterning

After the transfer of the graphene stacks, further graphical processing was performed on the two samples. Schematic diagrams of the graphical method using Al as the intermediate layer and corresponding optical photos are shown in Figure 2a–d. The sample Al-Gr (with Al still on the surface at this point) underwent standard photolithography (the AZ 5214 photoresist was used) to prepare the photoresist mask patterns, as shown in Figure 2a,b. It is noteworthy that a special developer, JRD DEV-1800 (purchased from Suzhou GADNY Chemical Co., Ltd., Suzhou, China), was used for the development process, as it can slow down Al corrosion. An optical photo after development is shown in Figure 2b, where the Al intermediate layer and pattern appear intact without excessive corrosion caused by the developer. Subsequently, the Al intermediate layer was dry-etched using Cl_2_ and BCl_3_, followed by graphene etching using oxygen plasma, as shown in Figure 2c. Finally, acetone and an Al etchant (3:1 DI water/3038) were used to remove the remaining photoresist and Al intermediate layer to obtain the graphene patterns (Figure 2d), and no obvious contaminant residue was observed. In contrast, for the TP-Gr sample, we used a traditional photolithography method, where the photoresist directly contacted graphene, as shown in Figure 2e, and a small amount of contamination could be observed on graphene.

### 2.3. Characterization

We first characterized the surface of graphene on both types of samples. This process primarily involved scanning electron microscopy (SEM, using JEOL, JSM-6390) and atomic force microscopy (AFM, using an instrument by Bruker, Billerica, MA, USA, ICON, Dublin, Ireland). Following that, further analysis of surface defects and doping of graphene was conducted using confocal Raman spectroscopy (LabRAM HR, excitation wavelength, 532 nm). X-ray photoelectron spectroscopy (XPS) analysis was conducted on the residual pollutants on graphene surface (PHI 5000 VersaProbe II). Finally, electrical measurements were performed on the samples to determine Hall mobility, resistance, and resistance–temperature variations. Hall mobility was measured using a non-contact Hall effect measurement system (Semilab, LEI-3200, Budapest, Hungary), while resistance and resistance–temperature variations were measured using a probe station with a hot plate and a multimeter. It should be noted that the samples used for the Hall mobility measurements were placed on high-resistivity silicon substrates.

## 3. Results

### 3.1. Surface Analysis

Firstly, the surface morphology of the graphene surface of the samples was observed via SEM and AFM, as shown in Figure 3. Due to the use of Al as an intermediate layer, which isolated graphene from the organic polymers, the surface of the Al-Gr sample was extremely clean (Figure 3a). In contrast, noticeable white contaminants were observed on the TP-Gr sample, as depicted in Figure 3b. Furthermore, surface topography images of the samples were captured using AFM (Figure 3c,d), and their root-mean-square roughness (*R*_q_) and mean roughness (*R*_a_) were calculated for a quantitative analysis of flatness. The surface of the Al-Gr sample was very smooth, with *R*_q_ of only 1.90 nm. In contrast, the TP-Gr sample showed noticeable white columnar bodies, and *R*_q_ reached 3.04 nm. The AFM results indicated that solvents such as acetone have difficulty completely removing the organic polymers introduced during traditional processes from the surface of graphene. The residual contaminants will tightly adhere to the graphene surface, causing a significant increase in roughness. Obviously, the residual organic matter severely degrades the quality of graphene. In contrast, the Al intermediate layer perfectly protected graphene from PMMA and photoresist contamination, greatly enhancing the cleanliness and consistency of the graphene surface.

Subsequently, an analysis of the Raman spectra of the samples was conducted to ascertain their defects and doping states. Three types of samples were prepared: Al-Gr, which was obtained by the graphical process involving the Al dielectric layer; TP-Gr* which was obtained by the conventional transfer process; and TP-Gr, which was obtained by the conventional patterning process. Each sample underwent 10 measurements, and the Raman spectra are shown in Figure 4a. The D peak located at 1350 cm^−1^ in the spectrum is commonly used to characterize defects, and the ratio of its intensity to that of the G peak (located at 1580 cm^−1^), *I*_D_/*I*_G_, is often employed to quantify defects and their degree in graphene. The purple region in Figure 4b displays the boxplots of *I*_D_/*I*_G_, and the average values for the samples Al-Gr, TP-Gr*, and TP-Gr were 0.137, 0.086, and 0.112, respectively. The *I*_D_/*I*_G_ value for Al-Gr was only slightly higher than those for TP-Gr* and TP-Gr, indicating that the defects introduced by the Al intermediate layer process were limited. The generation of defects may be attributed to the vigorous formation of H_2_ bubbles during the bubbling separation of graphene and the removal of the residual Al intermediate layer through corrosion. Therefore, further research is needed to control the bubbling separation process and study the composition of Al etchants. For TP-Gr, the *I*_D_/*I*_G_ sharply increased due to the further residues of the photoresist. Additionally, hole and electron doping affect the 2D peak (located at 2700 cm^−1^). Specifically, doping leads to a decrease in the intensity of the 2D peak and to a blue shift (hole doping) or a red shift (electron doping) [21]. After graphical processing, organic residue contamination caused a significant shift and reduction in the intensity of the 2D peak for TP-Gr. Figure 4b depicts a boxplot of the *I*_2D_/*I*_G_ ratio (blue region) and the position of the 2D peak (gray region), providing a visual and quantitative representation of the doping states of the samples. Specifically, the average *I*_2D_/*I*_G_ ratio value for TP-Gr was only 1.003, significantly lower than those for the other samples (Al-Gr: 1.526; TP-Gr*: 1.887). Combined with the observed blue shift of the 2D peak (2691.26 cm^−1^) for TP-Gr, this strongly suggests the presence of significant hole doping effects. The residual organic polymers during the transfer and lithography processes, along with water and oxygen molecules in the air, induce strong hole doping in graphene [22,23]. In contrast, the 2D peak of Al-Gr located at 2684.15 cm^−1^ tended to exhibit a red shift compared to the 2D peaks of TP-Gr* (2684.39 cm^−1^) and TP-Gr, which could be due to the remaining Al.

In the traditional transfer and patterning processes, the main organic polymers that come into contact with graphene are PMMA and the photoresist (AZ 5214). The primary macromolecular component of the photoresist is a novolak resin [24,25]. We employed the first-principles and density functional theory (DFT) methods to investigate the combination of polymers and Al elements with graphene (the calculation method is shown in Appendix B). Figure 5a–c illustrates the adsorption models of graphene surfaces with PMMA, photoresist (novolak resin), and Al. In addition, the binding energies between graphene surface and each substance were obtained through calculations (Figure A1). The binding energy of graphene–Al was only −0.48 eV, much lower than the binding energies of graphene–PMMA and graphene–novolak (−1.62 eV and −2.13 eV). This implies that the binding of graphene with polymers is stronger than that with Al, making it more difficult to remove polymers after their adhesion to graphene. Previous studies showed that using Al as an intermediate layer during the photolithography process effectively prevents the attachment of residual photoresist, while the residual amount of Al is extremely low [26]. This indicates that using Al as an intermediate layer to isolate graphene from various organic polymers and thus obtain a cleaner and more consistent graphene surface is a feasible approach.

XPS is an effective method for analyzing the surface chemistry and chemical bonds of materials. We utilized XPS to perform compositional analysis on two prepared samples to identify surface residues on graphene, as depicted in Figure 6. In Figure 6a, the C1s spectrum reveals a prominent sp2 C-C peak near 284.5 eV, which is a characteristic feature of graphene, along with a minor C-O peak near 286 eV. Additionally, the C=O peak around 290 eV is associated with oxygen-containing organic compounds [27]. The TP-Gr sample exhibited a notable presence of C=O peaks and higher C-O peaks compared to the Al-Gr sample, further demonstrating the presence of organic polymers on the surface of graphene prepared by conventional methods, which can be effectively mitigated by using the Al intermediate layer process. Notably, in Figure 6b, the Al2p spectrum shows no distinct Al characteristic peaks for Al-Gr, indicating the complete removal of Al from the graphene surface. These XPS spectra are consistent with the aforementioned DFT calculation results.

### 3.2. Electrical Measurements

In this work, we conducted electrical measurements with devices fabricated using both the Al intermediate layer process and conventional techniques. A statistical analysis of the measurement data was then employed to assess the extent to which the proposed method enhanced device performance and consistency. Firstly, Hall mobility and carrier density of the samples were measured and statistically analyzed. Twenty samples of each type were prepared, and the measurements were conducted at room temperature. For each type of sample, 20 specimens were prepared, and the statistical results are shown in Figure 7. The average Hall mobility of TP-GR* prepared using the traditional transfer method was 1045.28 cm^2^/(V·s), whereas the average Hall mobility of the samples obtained after lithography (referred to as TP-Gr samples) decreased further to 762.93 cm^2^/(V·s), as shown in Figure 7a. This mobility was significantly lower than the mobility of the samples prepared using the Al intermediate layer, denoted as Al-Gr, which was 1180.04 cm^2^/Vs. Moreover, the consistency for TP-Gr was also lower, with a standard deviation of 127.58 cm^2^/(V·s), higher than those for Al-Gr (standard deviation of 79.88 cm^2^/(V·s)) and TP-Gr* (standard deviation of 81.72 cm^2^/(V·s)). This was attributed to the contamination of TP-Gr by organic polymers, especially photoresist residues, which caused graphene to be hole-doped under the influence of moisture in the air [9], leading to a decrease in graphene mobility and other performance metrics. Additionally, the amounts and positions of these organic polymer residues could not be controlled, resulting in varying degrees of hole doping, and thus, a significant decrease in graphene consistency. Furthermore, as shown in Figure 7b, compared to TP-Gr, the carrier density of Al-Gr decreased from 1.19 × 10^13^ cm^−2^ to 9.09 × 10^12^ cm^−2^, and the distribution became more concentrated. The samples prepared using the Al intermediate layer technique exhibited higher mobility and lower carrier density and showed a more uniform distribution. This suggests that graphene mobility and consistency can be improved by preventing the deposition of residues of organic polymers such as PMMA and photoresists.

Furthermore, we conducted resistance measurements and statistical analyses on four samples (64 units) of devices containing Al-Gr or TP-Gr using a probe station, excluding damaged areas where resistance could not be measured. The results are shown in Figure 8a (62 devices for Al-Gr and 59 devices for TP-Gr). In the histogram, the resistance of the Al-Gr device is concentrated within the range of 1~2 kΩ, with mean and standard deviation values of 2.07 kΩ and 1.293 kΩ, respectively. Additionally, its normal distribution curve appears steeper and closer to the y-axis. In contrast, the resistance of the TP-Gr device is concentrated in the 2~3.5 kΩ range, with a higher average (3.66 kΩ) and standard deviation (1.887 kΩ) compared to those of the Al-Gr device, showing a smoother normal distribution curve. The statistical results indicated that the average resistance of the Al-Gr device was reduced by 43.44% compared to that of the TP-Gr device, with a decrease in standard deviation of 31.48%. The statistical data of device resistance and the normal distribution curve once again indicated that devices fabricated using the Al intermediate layer method exhibited lower and more concentrated resistances.

To further demonstrate the superiority of the Al intermediate layer process over traditional photolithography, resistance–temperature tests were conducted on temperature sensors fabricated using both processes. We conducted measurements of the resistance variation in the temperature sensors across the temperature range from 25 °C to 200 °C, with increments set at 25 °C intervals. Following each temperature adjustment, a waiting period of 10 min was observed to ensure the stabilization of the hot plate temperature before the measurements were performed. Figure 8b presents the resistance–temperature test results for the Al-Gr and TP-Gr devices. It can be observed that the resistance of graphene increased with temperature, exhibiting semiconductor characteristics. This was attributed to the dominance of phonon scattering in this temperature range, which increases with temperature, leading to a decrease in mobility [28,29]. The sensitivity of the Al-Gr and TP-Gr devices was calculated to be *S*_Al-Gr_ = 13.02 Ω/°C and *S*_TP-Gr_ = 7.79 Ω/°C, respectively. This indicated that the Al-Gr device possesses higher temperature sensitivity, which is 67.14% greater than that of the TP-Gr device. Performing linear regression on the resistance–temperature test curve yielded the goodness of fit (*R*^2^), reflecting the linearity of the relationship. The Al-Gr device exhibited a higher *R*^2^ value of 0.99408 compared to the TP-Gr device (*R*^2^ = 0.98748), indicating better linearity. Moreover, the shorter error bars for the Al-Gr device compared to the TP-Gr device indicated a more concentrated resistance distribution and better consistency for the former. It can be concluded that the Al-Gr device exhibited better sensitivity and consistency compared to the TP-Gr device. This is because graphene with lower mobility exhibits a weaker dependency of resistivity on temperature [30], which implies that the sensitivity of low-mobility graphene temperature sensors decreases. The Al intermediate layer process was shown to avoid the decrease in mobility caused by organic polymer residues and improve the cleanliness and consistency of the graphene surface, thereby enhancing sensor sensitivity and other performance metrics.

## 4. Conclusions

In the attempt to solve the issue of organic residue on graphene surfaces during graphene transfer and patterning processes using polymer materials, this study employed the deposition of Al as an intermediate layer to isolate graphene from PMMA support layers and photoresists. Through DFT simulations and calculations, it was determined that the binding energy between Al and the graphene surface was remarkably lower than that between PMMA or photoresist and graphene, indicating that Al is more easily removed from graphene, which makes it an excellent material for protecting graphene, when used as an intermediate layer. In both morphological images and Raman spectra, the samples prepared using the Al intermediate layer process exhibited cleaner and smoother surfaces with higher quality, practically preventing the deposition of residues of polymers and causing minimal additional defects. The XPS spectrum analysis similarly corroborated this assertion. Additionally, electrical measurements confirmed that the samples fabricated with the Al intermediate layer process had higher mobility, conductivity, and consistency, as evidenced by the statistical plots of Hall mobility and resistance. Furthermore, devices prepared using this method also demonstrated higher sensitivity, as shown by the resistance–temperature curve. Therefore, all results indicate that the Al intermediate layer process is highly beneficial for high-performance graphene devices. In the large-scale ultra-clean transfer and patterning production of graphene, the Al intermediate layer process will play a crucial role due to its advantages such as low cost and high technological maturity.

## Figures and Tables

**Figure 1 nanomaterials-14-00568-f001:**
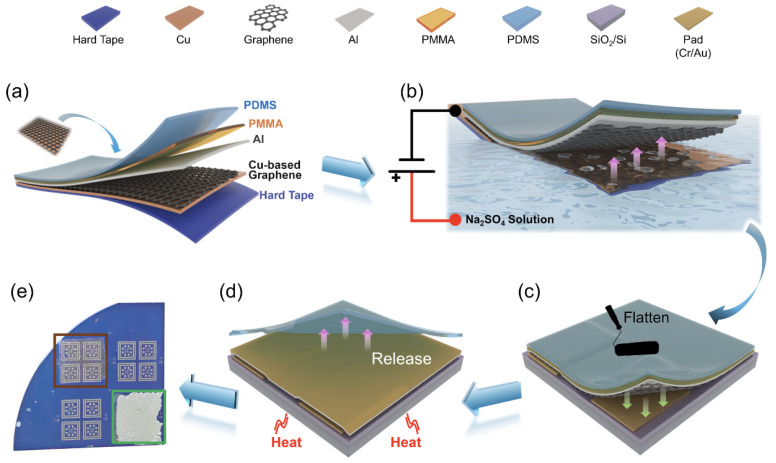
Schematic diagram of the graphene transfer method using an Al intermediate layer. (**a**) Preparation of PDMS/PMMA/Al/Gr/Cu/hard tape stack; (**b**) separation of graphene from the Cu substrate using the bubble separation method; (**c**) transfer of graphene onto the substrate; (**d**) thermal release of PDMS; (**e**) photograph of the chips after the transfer process (regions after Al intermediate-layer transfer and traditional dry transfer are indicated by green and brown boxes, respectively).

**Figure 2 nanomaterials-14-00568-f002:**
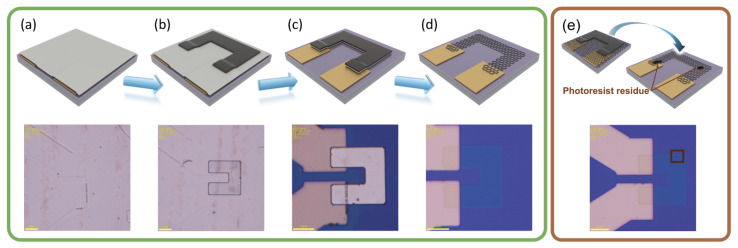
Graphene patterning via an Al intermediate layer. (**a**) Sample after Al intermediate layer transfer; (**b**) spin-coating of the photoresist followed by photolithography and development; (**c**) etching of the Al intermediate layer using BCl_3_ and Cl_2_, followed by oxygen plasma etching of graphene; (**d**) removal of the photoresist using acetone and of the Al intermediate layer using an Al etchant. Traditional patterning method: (**e**) direct spin-coating of the photoresist on graphene followed by patterning.

**Figure 3 nanomaterials-14-00568-f003:**
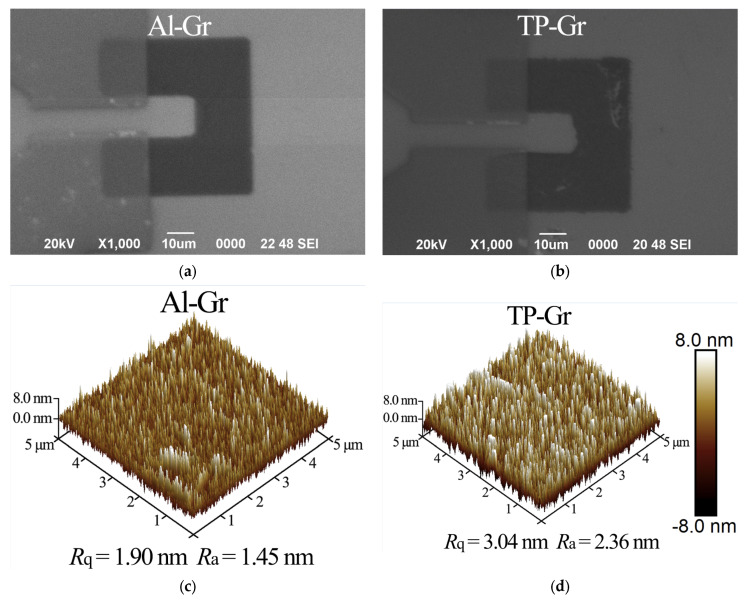
Images of graphene surface morphology. (**a**,**b**) Captured via SEM; (**c**,**d**) AFM three-dimensional surface morphology images (5 μm × 5 μm).

**Figure 4 nanomaterials-14-00568-f004:**
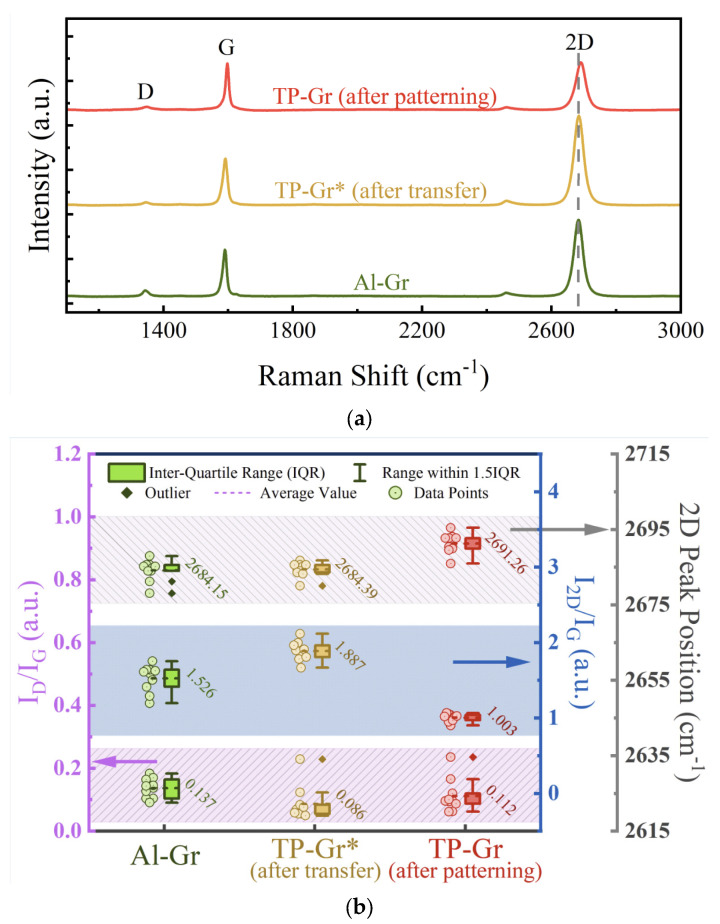
Raman test results. (**a**) Raman spectrum; (**b**) boxplots of *I*_D_/*I*_G_, *I*_2D_/*I*_G_, and 2D peak position.

**Figure 5 nanomaterials-14-00568-f005:**
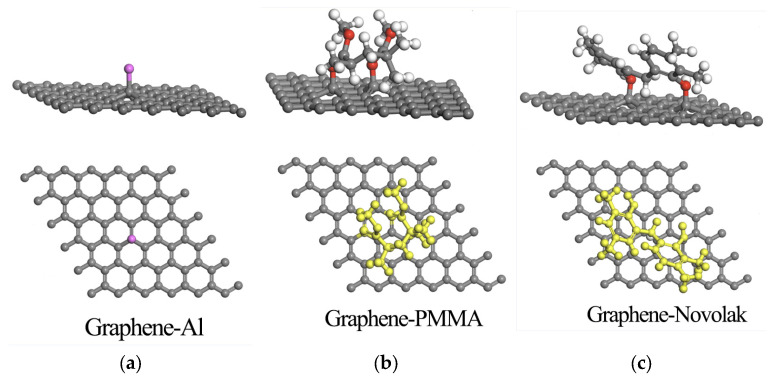
DFT optimized adsorption model of graphene with Al (**a**), PMMA (**b**), and novolak (**c**).

**Figure 6 nanomaterials-14-00568-f006:**
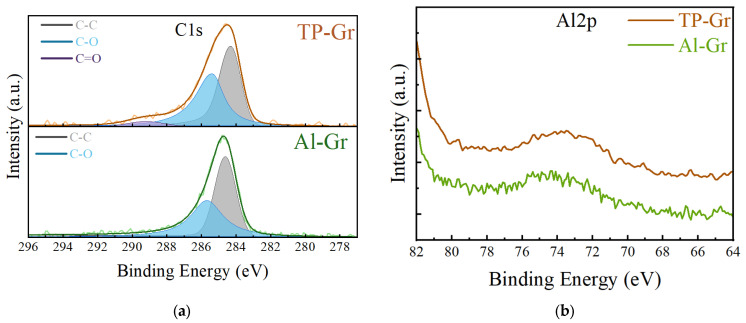
XPS spectrum for (**a**) C1s and (**b**) Al2p of two samples.

**Figure 7 nanomaterials-14-00568-f007:**
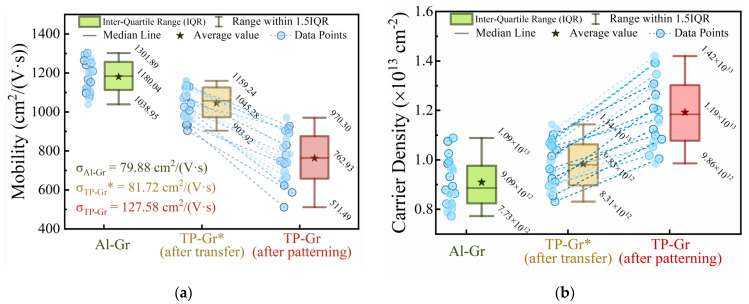
Boxplot of (**a**) Hall mobility and (**b**) carrier density.

**Figure 8 nanomaterials-14-00568-f008:**
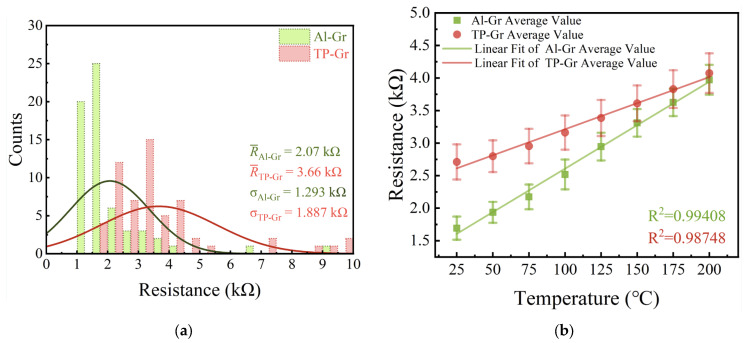
Electrical measurements. (**a**) Resistance statistics chart; (**b**) resistance–temperature test results.

## Data Availability

Data are contained within the article and Appendix A.

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
