# Peer review of "Enhancing the Consistency and Performance of Graphene-Based Devices via Al Intermediate-Layer-Assisted Transfer and Patterning"

_nanomaterials, 2024, doi:10.3390/nano14070568_

Round 1

Reviewer 1 Report

Comments and Suggestions for Authors

This work develops an approach of transferring and patterning CVD graphene using a Al intermediate layer which reduces the contamination and damage to graphene hence Enhancing the consistency and performance of graphene-based devices. Overall this is a solid work and the results are convincing. I only have two comments/suggestions.

1.     It helps to present the gate voltage dependence of the devices and compare between devices made with and without the Al intermediate layer.

2.     Figure 7 plots the histogram of device resistance. However since the resistance is affected by both carrier density and mobility, this plot doesn’t straightforwardly demonstrate the uniformity of the device characteristics. Instead, it is better to have two plots showing the histograms of the carrier density and mobility.

Reviewer 2 Report

Comments and Suggestions for Authors

1) Since the main goal of the work is to ensure that graphene is not chemically contaminated with PMMA, for which an intermediate layer in the form of aluminum was introduced, the authors need to experimentally demonstrate this result. For example, in the work cited by the authors [17], the XPS method was used for these purposes.
2) The authors need to expand the experimental part and add information standard for the CVD method (synthesis temperatures, gases used, gas flow, etc.)
3) Since the problem of transferring graphene synthesized by the CVD method is an extremely relevant topic, the authors should analyze and add links to current publications (no older than 5 years).

Author Response

请参阅附件。
